# Alignment Between Treatment Decision and Treatment Administration for Squamous Cell Carcinoma of the Upper Aerodigestive Tract Before, During, and After the COVID-19 Pandemic: A Retrospective Analysis

**DOI:** 10.3390/jcm14082613

**Published:** 2025-04-10

**Authors:** Benjamin Reliquet, Thomas Thibault, Paul Elhomsy, Dounia Chbihi, Mireille Folia, Caroline Guigou

**Affiliations:** 1Department of Otolaryngology-Head and Neck Surgery, Dijon University Hospital, 21000 Dijon, France; benjamin.reliquet@chu-dijon.fr (B.R.); mireille.folia@chu-dijon.fr (M.F.); 2Department of Internal Medicine, Dijon University Hospital, 21000 Dijon, France; thomas.thibault@chu-dijon.fr; 3Department of Anesthesiology and Critical Care, Dijon University Hospital, 21000 Dijon, France; paul.elhomsy@chu-dijon.fr; 4Department of Otolaryngology-Head and Neck Surgery, Croix Rousse Hospital, 69004 Lyon, France; douniachbihi@hotmail.com; 5ICMUB Laboratory, UMR CNRS 6302, University of Burgundy, 21000 Dijon, France

**Keywords:** professional practices, squamous cell carcinoma, multidisciplinary team meetings, upper aerodigestive tract, COVID-19 pandemic

## Abstract

**Objective**: The aim of this study was to assess the impact of the COVID-19 pandemic on the adequacy between treatment decisions made in multidisciplinary team meetings (MTMs) and therapy administered to patients with upper aerodigestive tract cancers. Secondary aims included assessing treatment administration times at different periods and identifying factors explaining discrepancies. **Methods**: A retrospective, monocentric study was conducted at a university hospital center from 2019 to 2021, including 475 first-line patients. Patients were divided into two groups: those with matching treatments (MTMs vs. delivered) and those with discrepancies. Alignment between treatment decision and treatment delivery was compared among the three periods (before, during, and after the COVID-19 pandemic), and factors influencing non-alignment were analyzed using univariate and multivariate analysis. **Results**: Of the 475 patients, 106 (23%) received treatments differing from MTM decisions. The pandemic period saw more advanced cancers (4.8% metastatic in 2019 vs. 12% in 2020), poorer general condition, and undernutrition. The pandemic did not significantly affect treatment matching (*p* = 0.4). Factors linked to mismatches included worse general condition (PS ≥ 2, *p* < 0.001) and more locally advanced tumors (T3/4, *p* = 0.002). Shorter processing times were noted during the pandemic and post-pandemic periods. **Conclusions**: Despite more advanced cancers and poorer general condition, patients treated during the pandemic had continuous care and similar treatment alignment as before. This study shows the effectiveness of ongoing care during the pandemic, ensuring treatment adherence.

## 1. Introduction

Squamous cell carcinoma accounts for 90% of histological types of tumors of the upper aerodigestive tract (UADT) [1]. With 500,000 new cases worldwide every year [2], this cancer is the fifth most common. In France, around 15,000 new cases are recorded every year [3]. Incidence is decreasing in men but has been increasing in women since the 1960s (+1.6%) [4].

A multidisciplinary team meeting (MTM) is an essential practice in the medical domain, particularly in cancer management [5]. This meeting brings together several healthcare professionals from different specialties (medical and paramedical) to draw up a personalized treatment plan for each patient presented. It is a legal requirement, as laid down in the French Public Health Code (article D. 6124-131).

In France, this MTM is part of a global roadmap called the cancer plan (created in 2003–2007) by the National Cancer Institute (INCa) [6], which aims to reduce the incidence of new cases and increase life expectancy and quality of life. The cancer plan focuses on prevention, screening, support, and research, as well as medical optimization and the reduction of inequalities.

While MTM recommendations reflect multidisciplinary consensus, the treating physician ultimately determines the final treatment plan, considering patient preferences, tolerability, comorbidities, and other relevant factors [7].

An initial study carried out by our team showed that for 19% of patients, the MTM decision did not match the treatment finally carried out [8]. This rate was like those found in the literature (between 15% and 22%) [9,10,11]. Treatment modifications, most often resulting in less aggressive therapy compared to the MTM recommendation, were primarily driven by a patient’s performance status (PS) ≥ 2, indicating a compromised general condition.

The COVID-19 pandemic necessitated a global healthcare system reorganization [12,13,14], impacting France significantly [15,16]. This disruption led to alterations in cancer treatment, including changes in surgical strategies (e.g., cancellations, reduced extents) and prolonged treatment timelines across specialties [17,18,19].

Building on our pre-pandemic (2019) study [8], we investigated the impact of the COVID-19 pandemic on treatment matching.

The primary aim was to assess the impact of the COVID-19 pandemic in the matching between therapeutic decisions made during multidisciplinary team meetings (MTMs) and the therapy administered to the patients with UADT. The secondary aims of this study were to assess the time required for diagnosis and treatment and to identify a change concerning factors influencing potential disparities.

## 2. Materials and Methods

### 2.1. Study Design

This study was a retrospective, monocentric, observational study conducted in a tertiary university hospital reference center, from 1 January 2019 to 31 December 2021.

The year 2019 was considered the pre-pandemic year and had already been the subject of previous work [8]. The year 2020 was considered the pandemic period. The year 2021 included the post-pandemic period after the appearance of the anti-COVID vaccine. Indeed, the first vaccine was marketed on 17 December 2020 [20]. From that date onwards, the COVID-19 pandemic was progressively considered to be “under control”.

Patients’ data was anonymized, and data processing was conducted in compliance with the MR-004 reference methodology of the National Commission on Informatics and Liberty (CNIL). Patients were informed. This study adhered to the principles of good clinical practice. This study was registered under the Health Data Hub number 20646009.

The inclusion criteria were as follows:

-Patients whose files were registered and discussed in MTMs at the university hospital center between 1 January 2019 and 31 December 2021.-Patients diagnosed initially with squamous cell carcinoma of the oral cavity, oropharynx, or hypopharynx or with cervical lymphadenopathy without a primary or larynx.-Patients for whom both the decided therapeutic plan and the executed therapeutic plan were recorded.-Patients in first-line treatment (initial diagnosis).

The exclusion criteria were as follows:

-Patients undergoing progressive disease or tumor recurrence.-Patients with tumor localization in the nasal cavities, sinuses, or salivary glands.-Patients with cancers that were categorized as cutaneous tumors.-Patients with anatomopathological types other than squamous cell carcinoma.-Patients with unknown treatment plans.-Patients whose records were presented in MTMs for expert consultation purposes.

### 2.2. Data Collected

The list of patients enrolled in MTMs was given by the Cancer Coordination Center of the tertiary referral center. Various data from the medical file were registered:-Patients’ characteristics (age, gender, World Health Organization Performance Status (PS) index [21]) and medical history (cirrhosis, diabetes, cardiovascular disease, chronic obstructive pulmonary disease (COPD)). Malnutrition status, history of cancer (current or remission), and alcohol and tobacco use were also recorded.-Squamous cell carcinoma characteristics: site of origin (oral cavity, oropharynx, larynx, hypopharynx, or cervical lymphadenopathy without primary), tumor node metastasis (TNM) status, according to the eighth edition of the UICC (Union for International Cancer Control) 2017 head and neck tumors TNM classification [22], and human papillomavirus (HPV) status.-Extension assessment, conducted in accordance with the recommendations of the French Society of Oto-Rhinolaryngology (SFORL), included panendoscopy of the UADT with an operative report and a summary diagram. Additionally, it involved a cervical–thoracic CT scan, a Positron Emission Tomography (PET-CT) scan, an Ear, Nose, and Throat (ENT) MRI, or an alternative ultrasound/scan-guided biopsy.-Multidisciplinary team meeting (MTM): MTMs were held once a week, jointly between the affiliated cancer center and the tertiary referral center. At least three physicians from the following specialties were involved: ENT, maxillofacial or reconstructive surgery, medical oncology, radiation oncology, pathology, and radiology. Before each meeting, a standardized form was completed and verified by the patient’s referring physician. The patient’s case was presented, and all aspects of the extension assessment were reviewed during the meeting. The form included the date, the attending physicians, and relevant prior data. After the meeting, the treatment protocol was collectively determined and validated by the MTM coordinator. The original form was stored in the Cancer Communication Folder (CCF), and a copy of the validated treatment protocol was added to the patient’s electronic medical record for traceability. Patients did not attend the MTM. Instead, the collective treatment decision was communicated to them during a disclosure consultation with their referring ENT physician, where they could either accept or decline the proposed treatment.-From the patient’s medical record, two time intervals were calculated: the time to care (time between the first symptom and specialist consultation), the time to treatment initiation (time from the MTM decision to the start of treatment).-Finally, the treatment center was documented (reference center or other center), as well as the 3-year survival.

Regarding the extension assessment, the completeness or incompleteness of the files when they were first presented to the MTM was noted. The work-up was considered complete when the file included pan endoscopy, anatomopathological results and a cervico-thoracic CT scan for laryngeal and hypopharyngeal localizations. MRI was added for oropharyngeal or oral cavity sites.

Examinations such as PET scans, ultrasound-guided biopsies, gastric or pulmonary fibroscopy were not considered as missing elements in the file, as they were requested on a case-by-case basis according to the patient.

### 2.3. Statistical Analysis

Microsoft Excel software (version 2022, Redmond, WA, USA) was used to collect data. R software (version 4.1.2., Miami, FL, USA) was used to perform statistical analyses.

Patient characteristics, extension assessments, tumor features, and treatment modalities are expressed as numbers and percentages for qualitative variables and as median and interquartile ranges for quantitative variables. Fisher’s exact test or Chi-squared test was used to compare qualitative variables, and Kruskal–Wallis test was used to compare quantitative variables. First, we compared variables according to the 3 study periods (pre-pandemic, pandemic and post-pandemic) on univariate analysis. Second, we performed univariate and multivariate analysis using logistic regression to highlight independent factors associated with inadequacy decision between the therapy delivered and the therapy decided in MTMs. The explanatory variables used were (1) patient characteristics, (2) tumor features, (3) extension assessments, and (4) study periods as defined before. Covariates with a *p*-value less than 0.2 in the univariate analysis were considered in the multivariate model [23]. A multivariate analysis was performed with a limited number of predictors to avoid overfitting [24]. The *p*-value < 0.05 was considered as significant.

## 3. Results

### 3.1. Patients Characteristics

One thousand two hundred and two patients received an MTM over a period from 1 January 2019 to 31 December 2021.

Of these patients, 475 were included in the study. The flowchart is detailed in Figure 1.

Fifty-five patients were excluded due to lack of information on their course and treatment received.

One hundred and thirty-two patients had recurrent disease (>6 months after the end of the first treatment) or ongoing disease (<6 months after the end of the first treatment) and were therefore excluded from this study.

Concerning tumor stages, 29 (7%) tumors were T0N+, 75 (15%) were T1, 90 (19%) were T2, 92 (19%) were T3, 189 (40%) were T4 over the three years combined.

Concerning tumor extension, 255 tumors (54%) were N ≥ 1 and 6% (N = 30) were metastatic at initial diagnosis. Tumors attributed to HPV infection accounted for 16% (N = 75) of lesions diagnosed.

The characteristics of the included population are summarized in Table 1.

Statistically significant differences were observed in both general condition (PS ≥ 2) and nutritional status among cancer patients when comparing pre-pandemic (2019) to pandemic/post-pandemic years (2020 and 2021). The proportion of patients with PS ≥ 2 increased from 8% in 2019 to 22% in 2020 and 28% in 2021 (*p* < 0.001). Similarly, the percentage of patients with poor nutritional status rose from 38% in 2019 to 74% in 2020 and 75% in 2021 (*p* < 0.001).

Medical file incompleteness at MTM presentation significantly increased during the pandemic (54% in 2020) and post-pandemic (39% in 2021) periods compared to pre-pandemic (23% in 2019) (*p* < 0.001). Similarly, the need for repeat MTM discussions before a therapeutic decision rose from 12% in 2019 to 37% in 2020 and 31% in 2021 (*p* < 0.001).

Despite no change in lymph node extension rates, metastatic cancer diagnoses significantly increased in 2020 (12%) compared to 2019 (4%) (*p* = 0.011).

### 3.2. Extension Assessment

As regards extension assessment, the completeness or incompleteness of files at the time of first presentation to the MTM was noted.

The characteristics of incomplete files are summarized in Table 2.

Incomplete medical files at the time of presentation to MTMs appeared to be more frequent during the pandemic period (54% in 2020) compared with the pre-pandemic period (23% in 2019) (*p* < 0.001). A higher number of MTMs passages for a single file was also noted in the 2020s (37% of files required a second passage to reach a decision, versus 12% in 2019 and 31% in 2021) (*p* < 0.001) (Table 1).

### 3.3. Treatment Performed

The treatments carried out in each year are summarized in Table 3.

### 3.4. Processing Time

We found no difference in the time taken to initially manage patients (time between the first consultation leading to a suspected diagnosis and referral to the MTM) (Table 4).

The time between MTM and the start of treatment during the pandemic period was shorter during the pandemic period (24 days) and post-pandemic period (26 days) than during the pre-pandemic period (32 days in 2019) (*p* < 0.001) (Table 4).

A patient who had undergone a total laryngectomy developed a COVID-19 infection on discharge from the ENT department. Adjuvant radiotherapy had to be delayed by 3 weeks due to the infection.

### 3.5. Inadequacy Between Treatment Decided in MTM and Treatment Delivered

In our overall population, 106 patients (23%) received a different treatment from that decided in MTM: 28 patients (19%) in 2019, 41 patients (25%) in 2020, and 37 patients (22%) in 2021 (Table 5).

Despite no change in lymph node extension rates, metastatic cancer diagnoses significantly increased in 2020 (12%) compared to 2019 (4%) (*p* = 0.011).

In these three periods, we found no differences according to the years or treatments initially decided on (*p* = 0.4). The trend for treatments not chosen by the MTM is towards de-escalation of treatments.

### 3.6. Factors Associated with Inadequacy Between Treatment Decided at MTM and Treatment Delivered

In univariate analysis (Table 6), the COVID-19 pandemic did not appear to have any influence on the mismatch between the treatment decided in MTM and that received by the patient (*p* = 0.4). Factors associated with mismatching versus matching appeared to be an altered general state defined by a PS ≥ 2 (42% of mismatching cases versus 13% of matching cases), a history of liver disease (13% of mismatching cases versus 5.7% of matching cases), a history of cardiovascular disease (66% of mismatching cases versus 53% of matching cases), undernutrition (83% of mismatching cases versus 57% of matching cases), chronic ethylism (54% of mismatching cases versus 39% of matching cases), and TNM status (81% of mismatching cases versus 53% of matching cases for tumor size >T2, 64% of mismatching cases versus 51% of matching cases for the presence of lymph node involvement, and 15% of mismatching cases versus 4.9% of matching cases for the presence of metastases).

In the multivariate analysis (Table 6), impaired general condition (PS ≥ 2, OR = 3.03 [95CI: 1.78–5.15], *p* < 0.001) and a tumor classified T3/4 (OR = 2.43 [1.40–4.35], *p* = 0.002) influenced the mismatch between treatment decided in MTM and that delivered to patients.

## 4. Discussion

The main objective of this study was to investigate the impact of the COVID-19 pandemic on the mismatch between treatments ordered in MTMs and those carried out. We found no change in the mismatch rate between the pre-pandemic year and the pandemic and post-pandemic years, with a stable rate of around 20%. Thus, the global pandemic did not influence the quality of management decisions at MTM for ENT cancer patients.

This 23% mismatch rate is externally consistent with the literature, whether specialized in ENT or not [9,10,11]. The tendency in cases of mismatch was towards therapeutic de-escalation, moving from curative to palliative treatment.

Patients with an altered general condition and a locally advanced tumor seemed to be those who would have a different treatment than that decided at the MTM. This has also been found in other studies [8]. Some authors have suggested that patients or one of their representatives (attending physician, family) should be present at the MTMs to manage the patient as a whole, and in particular his or her undernutrition status [25,26,27]. However, a study of 119 patients compared decisions made during ENT oncology MTMs before and after seeing and examining the patient and found a 97% matching rate between the two decisions [26].

Although the mismatch rate was not altered by the COVID-19 pandemic, patient characteristics differed between the pre-pandemic period and the pre- and post-pandemic periods, with patients presenting a more-worsened general condition, more significant undernutrition, and tumors more often diagnosed at a metastatic stage. This could indicate that patient access to services dedicated to the management of ENT tumors was made more difficult during the COVID-19 pandemic. Reduced access to care led to patients presenting with more advanced tumors, resulting in poorer general condition and worse prognoses.

The number of incomplete files has increased significantly, rising from 23% in 2019 to 54% in 2020, before falling back to 39% in 2021, with an increase in the number of MTM file passages without any impact on the final treatment decision at MTM. This could be explained by the difficulty patients have in obtaining appointments for complementary examinations during the pandemic and post-pandemic period.

Surprisingly, management times between the MTM decision and the start of treatment were shorter in the year 2020, the pandemic period. This could be explained by a reorganization of the healthcare system during the COVID-19 pandemic, with a reduction in functional management and greater availability of operating theatres for carcinologic management, which is consistent with another national study on the subject [28]. This would suggest that, within the ENT tumor management network, patient management is optimized. Prior studies have also demonstrated this adaptability of medical teams to maintain effective cancer care during the pandemic [28,29,30,31,32]. Optimization resulted from healthcare system reorganization, which prioritized cancer patients for operating shifts by rescheduling non-essential surgeries and consultations. A meta-analysis studied the impact of the COVID-19 pandemic on treatment delays for ENT cancers [33]. In this study, 24,898 patients were included in 19 articles. Seven studies (83.2% of the patients included) found, as in our work, a reduction in the time required to treat patients during the COVID-19 pandemic. In the various studies reviewed in this meta-analysis, several delays were studied, such as those between the first symptoms and biopsy or the first symptoms and diagnosis. As our treatment times are not the same (time between first consultation and MTM and time between MTM and treatment), they cannot be compared, even if our results are in the same direction as these studies (8-day-shorter time between MTM and initiation of treatment). Six other studies showed a stable time to treatment, and seven others an increase in this time during the COVID-19 period [33]. Only seven studies (representing 12.7% of patients included) found an increase in tumor stage. This suggests that the healthcare system is capable of adapting to a pandemic.

Consistent with national studies [28,29], MTM patient registrations remained stable during the pandemic (N = 179 in 2019, N = 212 in 2020, N = 206 in 2021 in our series).

In our work, we have described three distinct periods: pre-pandemic (2019), pandemic (2020), and post-pandemic with the beginning of vaccination (2021). Some authors had opted for a temporal segmentation using pandemic waves [28]. We have preferred to divide this period into three parts, as the resumption of “normal” hospital operations could only take place gradually with the arrival of the vaccine [34] and remained disrupted between pandemic waves [35,36,37].

In France, the MTM was instituted with the Cancer Plan, a public health strategy aimed at optimizing treatment that is regularly updated (the fourth version running from 2021 to 2030) [38]. It is endorsed and compulsory [39]. A French study carried out on several medical specialties showed that the MTM is perceived as legitimizing the medical decision and validating it from a medico-legal point of view [40,41]. In cases where a decision based on a reference framework can be applied, this decision is generally rapid and does not require in-depth discussion. For complex clinical cases (25–40%), a multidisciplinary approach improves decision-making [42]. A comprehensive summary document detailing pathology and treatment decisions serves as a crucial communication tool for patients and referring physicians [43].

Improving MTM efficiency may involve optimizing patient data collection, organization, and presentation. Several countries around the world are already using artificial intelligence to aid clinical decision support (CDS) in the field of oncology. These tools have already demonstrated their relevance [44,45,46,47], as well as the satisfaction they have generated among users [48]. They include, among other things, the collation of data provided by each specialist and the presentation of possible treatments at the MTM, using machine learning to compare them with available references and publications. In this way, treatments are classified into three categories: recommended treatments, treatments to be studied, and no-recommended treatments. To our knowledge, this system is not currently used in the field of ENT carcinology, nor in other medical specialties.

One of the limitations of our work lies in the collection method, which was spread over a period of more than 3 years. Data collection was performed by two individuals in 2019 but by only one individual during the pandemic and post-pandemic periods. This could have induced a collection bias.

This retrospective, single-center study presents inherent limitations. It would have been interesting to compare the evolution of ENT carcinology care during the COVID-19 pandemic in different tertiary referral centers, to see whether the results found in this work were homogeneous in France. Future research should expand to a national or European scale and compare the COVID-19 pandemic’s impact on relapsed/progressive versus first-line cancer treatment management. It could also have been interesting to study the impact of the COVID-19 pandemic on nasal cavity and paranasal sinuses cancers. This could have increased the population of this work even if it would have been more heterogeneous, with different anatomopathological results. This may be the subject of future work.

## 5. Conclusions

This study showed that the COVID-19 pandemic had no impact on the matching between treatments decided at the MTM and those administered, with an overall matching rate of 77%. The factors influencing mismatch between these two treatments were a deterioration in the patient’s general condition (PS ≥ 2) and a locally advanced tumor (T3/4). The trend in cases of mismatch was towards a reduction in treatment, from curative to palliative.

Patients treated during this pandemic period were more malnourished, in poorer general condition, and their tumors were more often metastatic, which may suggest greater difficulty in accessing care. On the other hand, treatment times seemed to have been shortened during the COVID-19 pandemic, demonstrating the healthcare system’s ability to adapt to severe pathologies during health crises.

## Figures and Tables

**Figure 1 jcm-14-02613-f001:**
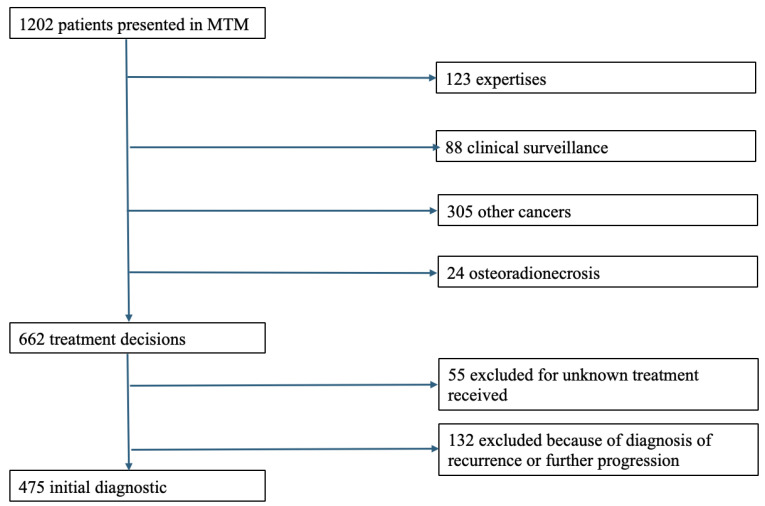
Flowchart.

**Table 1 jcm-14-02613-t001:** Population characteristics. Age and number of pack-years are expressed as median (Q1,Q3). COPD: chronic obstructive pulmonary disease; CV: cardiovascular; PY: pack-year; HPV: human papilloma virus; T: tumor; N: lymph node metastasis; M: distant metastasis; MTM: multidisciplinary team meeting.

Characteristics	2019 N = 147 ^1^	2020 N = 162 ^1^	2021 N = 166 ^1^	*p*-Value ^2^
Age	64 (56.72)	65 (59.73)	65 (59.73)	0.2
Male gender	117 (80%)	126 (78%)	127 (77%)	0.8
Performance status (≥2)	12 (8.2%)	36 (22%)	46 (28%)	**<0.001**
COPD	39 (27%)	32 (20%)	26 (16%)	0.061
Hepatopathy	11 (7.5%)	16 (9.9%)	8 (4.8%)	0.2
Cardiovascular disease	84 (57%)	87 (54%)	95 (57%)	0.8
Diabetes	16 (11%)	18 (11%)	30 (18%)	0.11
Undernutrition	56 (38%)	120 (74%)	124 (75%)	**<0.001**
Previous cancer history	26 (18%)	42 (26%)	32 (19%)	0.2
Smoke quantity (PY)	40 (15.50)	30 (10.45)	40 (20.50)	0.3
Active smoking	88 (60%)	89 (55%)	82 (49%)	0.2
Alcoholism	54 (37%)	69 (43%)	77 (46%)	0.2
HPV	22 (15%)	23 (14%)	25 (15%)	>0.9
T at diagnosis (3 or 4)	84 (57%)	99 (61%)	99 (60%)	0.8
N + at diagnosis	88 (60%)	87 (54%	80(48%)	0.12
M + at diagnosis	7 (4.8%)	20 (12%)	7 (4.2%)	**0.011**
Discovery of synchronous cancer	14 (9.5%)	16 (9.9%)	19 (11%)	0.9
Complete file	113 (77%)	75 (46%)	102 (61%)	**<0.001**
Adequacy MTM–treatment performed	119 (81%)	121 (75%)	127 (77%)	0.4
Place of treatment (university hospital or another center)	112 (81%)	134 (83%)	135 (81%)	0.5
Unknown	10	0	0	
MTM reconducted	17 (12%)	59 (37%)	51 (31%)	**<0.001**
Unknown	0	1	0	
Gross survival at 3 years	71 (48%)	89 (55%)	100 (60%)	0.11

^1^ n (%); Median (Q1,Q3). ^2^ Kruskal–Wallis rank sum test; Fisher’s exact test.

**Table 2 jcm-14-02613-t002:** Characteristics of incomplete files.

Diagnoses		2019 N = 147 (100%)	2020 N = 162 (100%)	2021 N = 166 (100%)
Number of incompletes (%)		34 (23)	87 (54)	64 (39)
Missing element in extension assessment	Imaging	26 (18)	67 (41)	47 (28)
	Anathomopathology		3 (2)	1 (1)
	Imaging + anathomopathology	2 (1)	2 (1)	4 (2)
	Pan endoscopy	6 (4)	8 (6)	7 (4)
	Pan endoscopy + imaging		7 (4)	5 (3)

**Table 3 jcm-14-02613-t003:** Treatment carried out for the entire study population.

	2019 N = 147 (100%)	2020 N = 162 (100%)	2021 N = 166 (100%)
Surgery	32 (22)	30 (19)	23 (14)
Exclusive radiotherapy	20 (14)	16 (10)	21 (13)
Radio-chemotherapy	40 (27)	49 (30)	49 (30)
Surgery and adjuvant radiotherapy	14 (10)	12 (7)	21 (13)
Surgery and radio-chemotherapy	20 (14)	16 (10)	26 (16)
Exclusive chemotherapy	6 (4)	23 (14)	14 (8)
No treatment performed	15 (10)	16 (10)	12 (7)

**Table 4 jcm-14-02613-t004:** Treatment times. Times are in days. Median (min, max). MTM: multidisciplinary team meeting.

Delay	2019 N = 147	2020 N = 162	2021 N = 166	*p*-Value ^1^
Initial care time (first consultation and MTM) in days	23 (16.31)	24 (16.40)	26 (17.38)	0.5
Unknown	0	2	0	
Time between MTM and treatment	32 (20.42)	24 (14.38)	26 (13.37)	**<0.001**
Unknown	15	18	14	

^1^ Kruskal–Wallis rank sum test.

**Table 5 jcm-14-02613-t005:** A cross-tabulation table outlining the treatments ultimately administered or patient outcomes in cases where the final treatment differed from the MTM decision at the time of initial diagnosis, in pre-pandemic period (2019) (A), in pandemic period (2020) (B), and in post-pandemic COVID-19 period (2021) (C). MTM: multidisciplinary team meeting; S.: surgery; RT.: radiotherapy; RTCT.: radio-chemotherapy; CT.: chemotherapy.

	Treatment	S.	RT.	S. + Adjuvant RT.	S. + Adjuvant RTCT.	Concomitant RTCT.	Palliative CT.	Supportive Care Treatment	Death Before Treatment	Treatment Refused by the Patient	Total
MTM Decision											
(**A**)											
S.		-	1	0	0	0	0	0	0	0	1
RT.		0	-	0	0	0	0	3	3	0	6
S. + adjuvant RT.		3	0	-	0	0	0	0	1	0	4
S. + adjuvant RTCT.		1	0	1	-	2	0	0	0	0	4
Concomitant RTCT.		1	2	0	1	-	1	2	2	0	9
Palliative CT.		0	0	0	0	1	-	0	2	1	4
Total		5	3	1	1	3	1	5	8	1	28
(**B**)											
S.		-	2	1	2	3	1	0	0	0	9
RT.		0	-	0	0	1	1	1	0	0	3
S. + adjuvant RT.		3	0	-	0	0	1	0	0	0	4
S. + adjuvant RTCT.		0	0	3	-	0	0	0	2	0	5
Concomitant RTCT.		1	1	0	0	-	4	1	3	1	11
Palliative CT.		1	0	0	0	1	-	1	6	0	9
Total		5	3	4	2	5	7	3	11	1	41
(**C**)											
S.		-	2	0	0	4	0	1	2	1	10
RT.		0	-	0	0	2	2	1	2	0	7
S. + adjuvant RT.		2	0	-	1	0	1	0	0	0	4
S. + adjuvant RTCT.		0	2	0	-	0	0	0	0	0	2
Concomitant RTCT.		0	1	0	1	-	6	0	2	0	10
Palliative CT.		0	0	0	0	1	-	0	3	0	4
Total		2	5	0	2	7	9	2	9	1	37

**Table 6 jcm-14-02613-t006:** Univariate and multivariate analysis according to adequacy between treatment decided in MTM and treatment finally delivered (mismatching versus matching group). PS: performance status; COPD: chronic obstructive pulmonary disease; T: tumor; N: lymph node metastasis; M: distant metastasis.

	Univariate Analysis			Multivariate Analysis		
Variables	Mismatching N = 108 ^1^	Matching N = 367 ^1^	*p*-Value ^2^	OR For No-Matching	95% CI	*p*-Value
PS ≥ 2	45 (42%)	49 (13%)	**<0.001**	3.03	1.78, 5.15	**<0.001**
COPD	27 (25%)	70 (19%)	0.2	1.03	0.57, 1.80	>0.9
Hepatopathy	14 (13%)	21 (5.7%)	**0.011**	1.59	0.68, 3.65	0.3
Cardiovascular disease	71 (66%)	195 (53%)	0.020	1.34	0.82, 2.23	0.2
Diabetes	17 (16%)	47 (13%)	0.4			
Undernutrition	90 (83%)	210 (57%)	**<0.001**	1.77	0.97, 3.36	0.07
Smoking (active or withdrawal)	50 (46%)	209 (57%)	0.051	0.85	0.50, 1.44	0.5
Alcoholism	58 (54%)	142 (39%)	**0.005**	1.11	0.64, 1.92	0.7
Previous cancer history	27 (25%)	73 (20%)	0.3			
T at diagnostic (3/4)	87 (81%)	195 (53%)	**<0.001**	2.43	1.40, 4.35	**0.002**
N at diagnostic (1/2/3+)	69 (64%)	186 (51%)	0.016	1.20	0.72, 1.99	0.5
M at diagnostic	16 (15%)	18 (4.9%)	**<0.001**	1.77	0.79, 3.92	0.2
Discovery of synchronous cancer	12 (11%)	37 (10%)	0.8			
Complete file	61 (56%)	229 (62%)	0.3			
Study period			0.4			
- 2019 (pre-pandemic)	28 (26%)	119 (32%)				
- 2020 (pandemic)	41 (38%)	121 (33%)				
- 2021 (post-pandemic)	39 (36%)	127 (35%)				

^1^ n (%). ^2^ Pearson’s Chi-squared test. OR: odds ratio. 95% CI: 95% confidence internal.

## Data Availability

The raw data supporting the conclusions of this article will be made available by the authors on request. All data are available on request by sending an e-mail to the corresponding author: caroline.guigou@chu-dijon.fr.

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
