# Peer review of "Alignment Between Treatment Decision and Treatment Administration for Squamous Cell Carcinoma of the Upper Aerodigestive Tract Before, During, and After the COVID-19 Pandemic: A Retrospective Analysis"

_jcm, 2025, doi:10.3390/jcm14082613_

Round 1

Reviewer 1 Report

Comments and Suggestions for Authors

The present study provides a picture of an intense and challenging period we have left behind not without consequences; it shows, among other things, the efficiency of the French healthcare system to consider priorities and to cope with a shortage of resources, since the time elapsed between the MTM decision and the start of treatment was surprisingly shorter in 2020 and did not lead to crucial delays.

It would have been interesting to include patients with nasal cavity and paranasal sinuses SCC, in order to examine in the paper all the upper aerodigestive tract subsites and rely on a higher number of individuals, which can be statistically important in a monocentric study.

Also, I would have considered in the inclusion criteria not only the first-line treatment patients, so to understand if, in case of persistence/recurrence of the disease, we could obtain more information about the discrepancy factors studied, possibly affected by the pandemic, if any. 

[line 45] Write "Multidisciplinary Team Meeting" to better introduce the acronym "MTM".

[line 106] Seek for the correct word: "Data" collected.

Author Response

Q.1 The present study provides a picture of an intense and challenging period we have left behind not  without consequences; it shows, among other things, the efficiency of the French healthcare system to consider priorities and to cope with a shortage of resources, since the time elapsed between the MTM decision and the start of treatment was surprisingly shorter in 2020 and did not lead to crucial delays.

A.1 Dear reviewer, thank you very much for your reply and for taking the time to review our work.

We hope that we have responded appropriately to each of your comments.

We remain available if necessary.

Best regards.

Q.2 It would have been interesting to include patients with nasal cavity and paranasal sinuses SCC, in order to examine in the paper all the upper aerodigestive tract subsites and rely on a higher number of individuals, which can be statistically important in a monocentric study.                                                                                                                                                                                                                                                                                                                                                     

A.2 Thank you for your comment.

It would indeed have been interesting to include patients with nasal cavity and paranasal sinuses cancers to increase the population included and to know what impact the COVID-19 pandemic had on their management. However, the anatomopathological characteristics of upper aerodigestive tract and nasal cavity and paranasal sinuses cancers are not the same.

This was added in the discussion as a limitation of our work (line 361): “It could also have been interesting to study the impact of the COVID-19 pandemic on nasal cavity and paranasal sinuses cancers. This could have increased the population of this work even if it would have been more heterogeneous, with different anatomopathological results. This may be the subject of future work”.

Q.3 Also, I would have considered in the inclusion criteria not only the first-line treatment patients, so to understand if, in case of persistence/recurrence of the disease, we could obtain more information about the discrepancy factors studied, possibly affected by the pandemic, if any. 

A.3 We couldn't agree more. In an initial study, we wanted to study only patients with first-line treatment.

We already have a goal to compare the impact of the COVID-19 pandemic on recurrence/recidive versus the first-line. This will be the subject of further work. This is already discussed in line 359.

Q.4 [line 45] Write "Multidisciplinary Team Meeting" to better introduce the acronym "MTM".

A.4 Now, the « Multidisciplinary Team Meeting » has been added to the text.

Q.5 [line 106] Seek for the correct word: "Data" collected.

A.5 Now, the « Data » has been added to the text.

Reviewer 2 Report

Comments and Suggestions for Authors

Review Report for Manuscript

"Matching Treatment Decisions for Squamous Cell Carcinoma of the Upper Aerodigestive Tract: A Retrospective Study in the Pre-, Per- and Post-pandemic COVID-19 Period"

This article presents data on treatment delivery during the COVID-19 period. The introduction, methodology, and results sections are well organized.

I suggest improving the presentation of inclusion and exclusion criteria. The exclusion criteria should be structured similarly to the inclusion criteria for better readability and consistency. Additionally, citations should be included for the World Health Organization Performance Status (PS) index and the 8th edition of the UICC (Union for International Cancer Control) staging system.

The duration of treatment is a crucial aspect in the COVID-19 period, as some patients experienced treatment delays due to infection. This is particularly relevant for patients undergoing radiotherapy or radiochemotherapy, where treatment interruptions could have impacted outcomes. It appears that this factor was not accounted for in the analysis, and I recommend incorporating this data to strengthen the study’s findings.

Furthermore, all abbreviations should be introduced below each table (e.g., COPD and MTM in Table 1).

My main concern is the timeframe in which treatment was delivered. These data should be included, as treatment delays could significantly affect outcomes in cancers of the upper aerodigestive tract.

Author Response

Q.1 "Matching Treatment Decisions for Squamous Cell Carcinoma of the Upper Aerodigestive Tract: A Retrospective Study in the Pre-, Per- and Post-pandemic COVID-19 Period"

This article presents data on treatment delivery during the COVID-19 period. The introduction, methodology, and results sections are well organized.

A.1 Dear reviewer, thank you very much for your reply and for taking the time to review our work.

We hope that we have responded appropriately to each of your comments.

We remain available if necessary.

Best regards.

Q.2 I suggest improving the presentation of inclusion and exclusion criteria. The exclusion criteria should be structured similarly to the inclusion criteria for better readability and consistency.

A.2 Thank you for your remark. The exclusion criteria are now presented in the same form as the inclusion criteria (line 100):

« The exclusion criteria were:

-           Patients undergoing progressive disease or tumor recurrence.

-           Patients with tumor localization in the nasal cavities, sinuses, or salivary glands.

-           Patients with cancers that were categorized as cutaneous tumors.

-           Patients with anatomopathological types other than squamous cell carcinoma.

-           Patients with unknown treatment plans.

-           Patients whose records were presented in MTMs for expert consultation purposes. »

Q.3 Additionally, citations should be included for the World Health Organization Performance Status (PS) index and the 8th edition of the UICC (Union for International Cancer Control) staging system.

A.3 These 2 references have now been cited and added to the text:

  1. Chow, R.; Chiu, N.; Bruera, E.; Krishnan, M.; Chiu, L.; Lam, H.; DeAngelis, C.; Pulenzas, N.; Vuong, S.; Chow, E. Inter-rater reliability in performance status assessment among health care professionals: a systematic review. Ann Palliat Med. 2016 Apr;5(2):83-92. doi: 10.21037/apm.2016.03.02. PMID: 27121736.
  2. Huang, S.H.; O'Sullivan, B. Overview of the 8th Edition TNM Classification for Head and Neck Cancer. Curr Treat Options Oncol. 2017 Jul;18(7):40. doi: 10.1007/s11864-017-0484-y. PMID: 28555375.

Q.4 The duration of treatment is a crucial aspect in the COVID-19 period, as some patients experienced treatment delays due to infection. This is particularly relevant for patients undergoing radiotherapy or radiochemotherapy, where treatment interruptions could have impacted outcomes. It appears that this factor was not accounted for in the analysis, and I recommend incorporating this data to strengthen the study’s findings.

A.4 Thank you very much for your comment.

In our series, a patient undergoing total laryngectomy developed a COVID-19 infection on discharge from hospital. His adjuvant radiotherapy had to be delayed by 3 weeks.

This has now been added to the text as follows (line 227):

“A patient who had undergone a total laryngectomy developed a COVID-19 infection on discharge from the ENT department. Adjuvant radiotherapy had to be delayed by 3 weeks due to the infection.”

Q.5 Furthermore, all abbreviations should be introduced below each table (e.g., COPD and MTM in Table 1).

A.5 All abbreviations are now introduced in the various tables.

Q.6 My main concern is the timeframe in which treatment was delivered. These data should be included, as treatment delays could significantly affect outcomes in cancers of the upper aerodigestive tract.

A.6 Thank you for your comment. However, we don't fully understand your question.

In fact, in the results section, there is a sub-section entitled “3.4. Processing time”, in which we show that there was no delay in taking charge during the COVID-19 period (line 220).

The results are summarized in table 4.  The time between MTM and the start of treatment during the pandemic period was shorter during the pandemic period (24 days) and post-pandemic period (26 days) than during the pre-pandemic period (32 days in 2019) (p<0.001).

Should you require any further information, please do not hesitate to contact us.

Reviewer 3 Report

Comments and Suggestions for Authors

The topic of the alignment between the treatment decided and the treatment administered during the COVID19 pandemic regarding squamous cell carcinoma of the upper aerodigestive tract is interesting and is not sufficiently represented in literature.

The manuscript is well written, the data is presented clear, and the statistical analysis shows sufficient knowledge, and the discussion is well considered.

The manuscript lacks a more rigorous read line. I would recommend excluding the analysis of time interval, since data of many patients is missing (table 4), a focus on non-alignment and its influencing factors seem to be more interesting, and the analysis of time delay has already been covered in the literature: Grumstrup Simonsen M, Fenger Carlander AL, Kronberg Jakobsen K, Grønhøj C, Von Buchwald C. The impact of the COVID-19 pandemic on time to treatment in head and neck cancer management: a systematic review. Acta Oncol. 2025 Jan 28;64:156-166. doi: 10.2340/1651-226X.2025.41366.

I recommend checking, evaluating, and eventually correcting the following items:

- Title proposal: Alignment between treatment decision and treatment administration for squamous cell carcinoma of the upper aerodigestive tract before, during, and after the COVID19 pandemic: a retrospective analysis (as I see no matching, and the term per-pandemic might be unusual?)

- Abstract methods: Focus on 1) alignment between treatment decision and treatment administration compared between 3 periods (before, during, and after COVID19 pandemic) and 2) factors influencing non-alignment (bivariable and multivariable analysis was performed)

- Abstract results: Concordance? Was that analyzed for example with Cohens-/Fleiss-Kappa?

- Text line 100: Patients with lesions THAT were categorized as cutaneous tumors.

- Text line 162: bi-/multivariable= 1 dependent variable (your situation) / bi-/multivariate= >1 dependent variable (not your situation)

- Text line 186: the sum is 474; is one patient missing?

- Table 1: Kruskal-Wallis test for testing 3 groups; no Mann-Whitney-test for testing each 2 groups?

- Table 4: Fishers-Exact test? The table shows only metric data (Kruskal-Wallis test applicable)

- Text line 330: Matching learning or Machine learning?

- Text line 355: AI should not be mentioned here, as it is not related to the study, and: non-alignment is caused by patient deterioration and locally advanced tumors, which is presumably not changeable by AI

Author Response

Q.1 The topic of the alignment between the treatment decided and the treatment administered during the COVID19 pandemic regarding squamous cell carcinoma of the upper aerodigestive tract is interesting and is not sufficiently represented in literature.

The manuscript is well written, the data is presented clear, and the statistical analysis shows sufficient knowledge, and the discussion is well considered.

A.1 Dear reviewer, thank you very much for your reply and for taking the time to review our work.

We hope that we have responded appropriately to each of your comments.

We remain available if necessary.

Best regards.

Q.2 The manuscript lacks a more rigorous read line. I would recommend excluding the analysis of time interval, since data of many patients is missing (table 4)

A.2 Thank you very much for your comment. The last line of table 4 has been deleted (the time between first and adjuvant treatment) as you suggested.

We would still like to keep this table which shows the absence of delays in care during the COVID-19 period and therefore the adaptability of the care system during the pandemic period.

Q.3 A focus on non-alignment and its influencing factors seem to be more interesting, and the analysis of time delay has already been covered in the literature: Grumstrup Simonsen M, Fenger Carlander AL, Kronberg Jakobsen K, Grønhøj C, Von Buchwald C. The impact of the COVID-19 pandemic on time to treatment in head and neck cancer management: a systematic review. Acta Oncol. 2025 Jan 28;64:156-166. doi: 10.2340/1651-226X.2025.41366.

A.3 Thanks for the recent reference we hadn't seen. From now on, the reference is cited in the discussion as follows (line 311): « A meta-analysis studied the impact of the COVID-19 pandemic on treatment delays for ENT cancers [33]. In this study, 24,898 patients were included in 19 articles. Seven studies (83.2% of the patients included) found, as in our work, a reduction in the time required to treat patients during the COVID-19 pandemic.  In the various studies reviewed in this meta-analysis, several delays were studied, such as those between the first symptoms and biopsy, or the first symptoms and diagnosis. As our treatment times are not the same (time between first consultation and MTM, and time between MTM and treatment), they cannot be compared, even if our results are in the same direction as these studies (8-day shorter time between MTM and initiation of treatment). Six other studies showed a stable time to treatment, and 7 others an increase in this time during the COVID-19 period [33]. Only 7 studies (representing 12.7% of patients included) found an increase in tumor stage. This suggests that the healthcare system is capable of adapting to a pandemic.”

I recommend checking, evaluating, and eventually correcting the following items:

Q.4 Title proposal: Alignment between treatment decision and treatment administration for squamous cell carcinoma of the upper aerodigestive tract before, during, and after the COVID19 pandemic: a retrospective analysis (as I see no matching, and the term per-pandemic might be unusual?)

A.4 Thank you very much for your comment. The title has been changed in line with your recommendations.

Q.5 Abstract methods: Focus on 1) alignment between treatment decision and treatment administration compared between 3 periods (before, during, and after COVID19 pandemic) and 2) factors influencing non-alignment (bivariable and multivariable analysis was performed)

A.5 Now, the abstract methods has been modified as follow (line 22):

« A retrospective, monocentric study was conducted at a university hospital center from 2019 to 2021, including 475 first-line patients. Patients were divided into two groups: those with matching treatments (MTMs vs. delivered) and those with discrepancies. Alignment between treatment decision and treatment delivery was compared between the 3 periods (before, during and after the COVID-19 pandemic), and factors influencing non-alignment were analyzed using univariate and multivariate analysis.»

Q.6 Abstract results: Concordance? Was that analyzed for example with Cohens-/Fleiss-Kappa?

A.6 Sorry for the mistake. Now, “concordance” has been changed by “matching”.

Q.7 Text line 100: Patients with lesions THAT were categorized as cutaneous tumors.

A.7 Now, this sentence has been modified.

Q.8 Text line 162: bi-/multivariable= 1 dependent variable (your situation) / bi-/multivariate= >1 dependent variable (not your situation)

A.8 Thank you for your comment. to reduce confusion, the term “bivariate” has been changed throughout the text to “univariate”.

Q.9 Text line 186: the sum is 474; is one patient missing?

A.9 Thank you for your comment. One patient was missing in the T0 group.

Now, the sentence has been modified as follow (line 183):

“Concerning tumor stages, 29 (7%) tumors were T0N+, 75 (15%) were T1, 90 (19%) were T2, 92 (19%) were T3, 189 (40%) were T4 over the three years combined.”

Q.10 Table 1: Kruskal-Wallis test for testing 3 groups; no Mann-Whitney-test for testing each 2 groups?

A.10 Thank you very much for your reply. We have indeed chosen not to carry out a Mann-Withney test for table 1 because we are comparing 3 independent groups (the 3 periods). We therefore used a Kruskal-Wallis test to test the overall difference between the 3 groups.

We could have compared each group two by two using a Mann-Withney test, but this would not have been very useful, and would have led to an inflation of the alpha risk by carrying out 3 additional tests.

Our statistical tests were carried out by Dr. Thomas Thibault, a specialist in statistics. We remain at your disposal if you would like us to carry out an additional Mann-Withney test for the results in Table 1.

Q.11 Table 4: Fishers-Exact test? The table shows only metric data (Kruskal-Wallis test applicable)

A.11 Sorry for the confusion. The term “Fishers-Exact Test” has been removed.

Q.12 Text line 330: Matching learning or Machine learning?

A.12 Sorry for the confusion. « Maching-learning » has been added to the text.

Q.13 Text line 355: AI should not be mentioned here, as it is not related to the study, and: non-alignment is caused by patient deterioration and locally advanced tumors, which is presumably not changeable by AI.

A.13 This sentence has been removed from the conclusion following your recommendations.

Round 2

Reviewer 2 Report

Comments and Suggestions for Authors

The authors answered all the issues.

Reviewer 3 Report

Comments and Suggestions for Authors

Dear authors,

First of all, I would like to thank you again for the opportunity to review the manuscript.
The authors have critically evaluated the comments of the previous review and, taking into account several reviewers with different opinions, have implemented them satisfactorily as far as possible. 
Two comments: 
A) Q.8 Text line 162: bi-/multivariable= 1 dependent variable (your situation) / bi-/multivariate= >1 dependent variable (not your situation) - A.8 Thank you for your comment. to reduce confusion, the term “bivariate” has been changed throughout the text to “univariate”.

I meant that a dependent variable (here the treatment deviation) is generally referred to as a multivariate regression and not a multivariate regression. However, it should be borne in mind that the two terms are often used interchangeably in the literature.

B) Q.12 Text line 330: Matching learning or machine learning? - A.12 Sorry for the confusion. " Maching-learning " has been added to the text

I meant that you should write MachinE Learning here.

Thanks for the effort!